

# Impact of 2020 COVID-19 lockdowns on particulate air pollution across Europe

Jean-Philippe Putaud[1], Enrico Pisoni[1], Alexander Mangold[2], Christoph Hueglin[3], Jean Sciare[4], Michael Pikridas[4], Chrysanthos Savvides[5], Jakub Ondracek[6], Saliou Mbengue[7], Alfred Wiedensohler[8], Kay Weinhold[8], Maik Merkel[8], Laurent Poulain[8], Dominik van Pinxteren[8], Hartmut Herrmann[8], Andreas Massling[9], Claus Nordstroem[9], Andrés Alastuey[10], Cristina Reche[10], Noemí Pérez[10], Sonia Castillo[11], Mar Sorribas[12], Jose Antonio Adame[12], Tuukka Petaja[13], Katrianne Lehtipalo[13], Jarkko Niemi[14], Véronique Riffault[15], Joel F. de Brito[15], Augustin Colette[16], Olivier Favez[16], Jean-Eudes Petit[17], Valérie Gros[17], Maria I. Gini[18], Stergios Vratolis[18], Konstantinos Eleftheriadis[18], Evangelia Diapouli[18], Hugo Denier van der Gon[19], Karl Espen Yttri[20], Wenche Aas[20].

[1]European Commission, Joint Research Centre (JRC), Ispra, Italy.
[2]Scientific Service Observations, Royal Meteorological Institute of Belgium, Brussels, Belgium.
[3]Swiss Federal Laboratories for Materials Science and Technology (EMPA), Duebendorf, Switzerland.
[4]Climate and Atmosphere Research Center, The Cyprus Institute, Nicosia, 2121, Cyprus.
[5]Ministry of Labour and Social Insurance, Department of Labour Inspection (DLI), Nicosia, Cyprus.
[6]Institute of Chemical Process Fundamentals, Czech Academy of Sciences, Prague, Czech Republic.
[7]Global Change Research Institute, Czech Academy of Sciences, Brno, Czech Republic.
[8]Atmospheric Chemistry Department (ACD), Leibniz Institute for Tropospheric Research (TROPOS), Leipzig, Germany.
[9]Department of Environmental Science, Aarhus University, Roskilde, Denmark.
[10]Institute of Environmental Assessment and Water Research (IDAEA-CSIC), Barcelona, 08034, Spain.
[11]Andalusian Institute for Earth System Research (IISTA-CEAMA), University of Granada, Granada, 18006, Spain.
[12]National Institute for Aerospace Technology (INTA), Mazaagón, Huelva, Spain.
[13]Institute for Atmospheric and Earth System Research INAR / Physics, University of Helsinki, Helsinki, Finland.
[14]Helsinki Region Environmental Services Authority (HSY), Helsinki, Finland.
[15]IMT Nord Europe, Institut Mines-Télécom, Univ. Lille, Centre for Energy and Environment, F-59000 Lille, France.
[16]Institut National de l'Environnement Industriel et des Risques, Verneuil-en-Hallate, France.
[17]Laboratoire des Sciences du Climat et de l'Environnement, Gif-sur-Yvette, France.
[18]Environmental Radioactivity & Aerosol Technology for Atmospheric & Climate Impact Lab, N.C.S.R. "Demokritos", 15341 Ag. Paraskevi, Attiki, Greece.
[19]TNO, Dept. Climate, Air and Sustainability, Utrecht, The Netherlands.
[20]NILU – Norwegian Institute for Air Research, P.O. Box 100, 2027 Kjeller, Norway.

*Correspondence to*: J.P. Putaud (jean.putaud@ec.europa.eu)



**Abstract.** To fight against the first wave of Coronavirus disease 2019 (COVID-19) in 2020, lockdown measures were implemented in most European countries. These lockdowns had well-documented effects on human mobility. We assessed the impact of the lockdown implementation and relaxation on air pollution by comparing daily particulate matter (PM), nitrogen dioxide ($NO_2$), and ozone ($O_3$) concentrations, as well as particle number size distributions (PNSD) and particle light absorption coefficients in-situ measurement data with values expected if no COVID-19 epidemic had occurred at 28 sites across Europe for the period 17 February – 31 May 2020. Expected PM, $NO_2$ and $O_3$ concentrations were calculated from the 2020 Copernicus Atmospheric Monitoring Service (CAMS) Ensemble forecasts, combined with 2019 CAMS Ensemble forecasts and measurement data. On average, lockdown implementations did not lead to a decrease in $PM_{2.5}$ mass concentrations at urban sites, while relaxations resulted in a $+26 \pm 21\%$ rebound. The impacts of lockdown implementation and relaxation on $NO_2$ concentrations were more consistent ($-29 \pm 17$ %, and $+31 \pm 30$ %, respectively). The implementation of the lockdown measures also induced statistically significant increases in $O_3$ concentrations at half of all sites ($+13$ % on average). An enhanced oxidizing capacity of the atmosphere could have boosted the production of secondary aerosol at those places. Changes in the wavelength dependence of the particle light absorption coefficients and PNSD were also examined at 14 and 13 sites, respectively. Since these variables are not calculated by the CAMS model, expected values were estimated from 2017 – 2019 measurement data. A significant change in the relative contributions of wood and fossil fuel burning to the concentration of black carbon during the lockdown was detected at 7 sites. The contribution of particles smaller than 70 nm to the total number of particles significantly changed at most of the urban sites, with a mean decrease of $-7 \pm 5$ % coinciding with the lockdown implementation. Our study shows that the response of $PM_{2.5}$ and $PM_{10}$ mass concentrations to lockdown measures was not systematic at various sites across Europe for multiple reasons, the relationship between road traffic intensity and particulate air pollution being more complex than expected.

## 1 Introduction

The first case of COVID-19 (Coronavirus disease 2019) in Europe was identified in Italy on 21 February 2020, although recent evidence suggests that the virus had already spread across northern Italy by mid-January (Cerqua and Di Stefano, 2022). National authorities took measures to limit the epidemics propagation across Europe and lockdown measures entered into force in various countries from March 2020. These measures led to dramatic decreases in activities such as road traffic (IEA, 2020), and large reductions in air pollutant emissions from these pollution sources were expected. Shortly after the first lockdown measures were implemented, numerous articles unsurprisingly reported about marked improvements in air quality across Europe (see examples in Putaud et al., 2021). These statements were mostly based on simple comparisons between 2020 and previous year data obtained from remote sensing or in-situ observations. Nonetheless, it was quickly shown that the impacts of the lockdown measures on air pollution were quite complex and could not be assessed without implementing sufficiently developed methodologies (Copernicus, 2020; Kroll et al., 2020; Shi et al., 2021; see also quotations in Schiermeier, 2020), including "deweathering" techniques (e.g. Goldberg et al., 2020; Petetin et al., 2020; Venter et al., 2020; Grange et al., 2021; Petit et al., 2021), modelling (Hammer et al., 2021; Yang et al., 2021) or combinations of model and measurement data (Le et al., 2020; Barré et al., 2021; Beloconi et al, 2021; Jiang et al., 2021). The latter were also applied by Putaud et al. (2021) to northern Italy, one of the most polluted areas in Europe, where the first major COVID-19 outbreak occurred in Europe. That work is extended here to about 30 urban and regional background sites across Europe, for which daily in-situ measurement data from February to May 2020 are compared to expected data (as if no COVID-19 epidemics had occurred) across the same period. Lockdown impacts on gas and particulate air pollution are thus statistically assessed and estimated. This analysis is complemented by the study of variations in intensive variables calculated from particle number size distribution (PNSD) and multi-wavelength particle light absorption coefficient measurement data. Since these variables are not produced by the model we used, measurements performed during previous years were in these cases used to determine the expected data.

## 2 Material and methods

This study focuses on the COVID-19 lockdowns that occurred across Europe in spring 2020. For the sake of clarity, the same three periods were considered for all countries: a 3-week-period before lockdowns were implemented (A, "ante", 17 February – 8 March 2020), a 6-week period for which mobility was minimal across Europe (D, "during", 23 March – 3 May 2020), and a 3-week-period during which lockdown measures were partially or totally relaxed (P, "post", 11 – 31 May 2020). Therefore, the 2 week period 9 – 22 March is excluded



from the analysis because lockdown measures were unevenly implemented across Europe at this time. Levels of stringency during periods A, D, and P in the various countries are discussed in Section 3.1 on the basis of mobility data.

Measurements of particulate matter ($PM_{10}$ and $PM_{2.5}$), nitrogen dioxide ($NO_2$), and ozone ($O_3$) surface level concentrations from 16 urban sites and 12 regional background sites located in 13 countries across Europe were examined for the three periods A, D and P. Measurement data from the same periods in 2019, together with model outputs for the same periods in both 2019 and 2020 were used to estimate the pollutant expected concentrations, which would have occurred in 2020 if no lockdown measures were applied. The potential impact of weather conditions on pollutant concentrations was therefore taken into account.

In addition to PM mass concentrations, two other variables characterising particulate air pollution were studied at 13 sites: (i) the Absorption Ångström exponent (AÅE), which describes the wavelength dependence of the particle light absorption coefficient and reflects the relative contributions of fossil fuel burning and wood burning to the atmospheric concentration of black carbon (Helin et al, 2021, and references therein), and (ii) the contribution of "small" particles ($N_{small}$) to the "total" number of particles ($N_{tot}$), as a proxy for primary particle emissions. Indeed, vehicle tail-pipe emissions have been shown to be dominated by particles whose mobility diameters ($D_p$) range between 15 and 70 nm (Giechaskiel et al, 2020; Garbariene et al., 2021). Wood combustion particle diameters are highly dependent on the combustion conditions. Particles with $D_p<70$ nm can be emitted by wood burners (Hueglin et al., 1997). We assumed that the number of particles larger than 15 nm is reasonably insensitive to new particle formation bursts (nucleation events). Both of the variables AÅE and $N_{small}/N_{tot}$ are intensive variables, i.e. they are not directly dependent on pollution dispersion and therefore much less sensitive to weather conditions than pollutant concentrations.

### 2.1 Mobility data

We could not find any statistical data whose time resolution was good enough (i.e. weekly or better) to assess lockdown impacts on human activities in a consistent way across all 13 countries considered in this study. Therefore, we focused on mobility data as proxies for lockdowns' stringencies. Driving route request data at city and regional scales temporarily made available by Apple® at covid19.apple.com/mobility (last accessed 21/03/2022) were used as an indicator of road-traffic intensity for all sites, except those in Cyprus for which such data were not available.

To assess the relationship between Apple® driving route request data and the actual number of vehicle kilometres driven, monthly motor fuel consumption from EUROSTAT (ec.europa.eu/eurostat) and AVENERGY (www.avenergy.ch) from January to May 2020 were used. For Cyprus, monthly activity data from the local authorities (www.cystat.gov.cy) were used.

### 2.2 Measurement sites

The 28 air pollution measurement sites considered in this study are shown in Figure 1. Details are listed in Table 1 where sites are sorted from North to South. Twenty-four of these sites constitute twin sites – 1 urban site and 1 regional background site in the same area (< 200 km).

### 2.3 Model data

We used CAMS (Copernicus Atmospheric Monitoring Service) Ensemble forecasts for $PM_{10}$, $PM_{2.5}$, $NO_2$, and $O_3$ daily surface level mass concentrations calculated as the median of the concentrations computed independently by nine different regional air quality models (Marecal et al., 2015), namely CHIMERE, DEHM, EMEP, EURAD-IM, GEM-AQ, LOTOS-EUROS, MATCH, MOCAGE, and SILAM. Each model is based on different schemes describing the formation, dispersion and deposition of reactive gases and particles, but uses the same meteorological fields from the ECMWF (European Centre for Medium-Range Weather Forecast) Integrated Forecasting System, and the same pollutant emission data derived from officially reported emissions for previous years, and therefore ignoring any potential lockdown effect (Denier van der Gon et al., 2015; Kuenen et al., 2022). The outputs of the nine individual models are interpolated on a common regular 0.1° x 0.1° latitude x longitude grid (about 10 km x 10 km) on 10 vertical levels from the surface layer (0-40 m) up to about 5 km altitude over Europe (defined as 25°W-45°E, 30°N-72°N). Median values are little sensitive to outliers (Riccio et al, 2007) and model ensembles are expected to yield better estimates than individual models (Galmarini et al., 2018).



**2.4 Measurement data**

2019 and 2020 $PM_{10}$, $PM_{2.5}$, $NO_2$, and $O_3$ measurement data from urban sites were collected from the local air quality monitoring networks, except for Athens, for which PM and $NO_2$ data originate from the ACTRIS (Aerosol,
Clouds and Trace gases Research Infrastructure) Observatory operated by the National Centre for Scientific Research "Demokritos". Measurement data from regional background sites were also all produced by ACTRIS Observatories operated by research performing organisations or EMEP (co-operative programme for monitoring and evaluation of the long range transmission of air pollutants in Europe) monitoring sites, and provided by the ACTRIS Data Centre. Pollutant concentrations were measured from 3 to 9 m above the ground with methods
listed in Tables S1 and S2 (Supplement).

PNSD and particle light absorption data from 2017 to 2020 originated from the authors' organisations. Data from ACTRIS sites were provided by the ACTRIS Data Centre, and data from other sites were specifically made available for this work. PNSD and particle light absorption coefficients were determined using instruments listed in Table S3.

**2.5 Data analysis**

Data were analysed as in Putaud et al., 2021. Briefly, 2020 expected daily concentrations ($Exp_{2020}$) were estimated from 2020 CAMS-Ensemble daily forecasts ($CAMS_{2020}$) and the ratio between 2019 daily observations ($Obs_{2019}$) and 2019 CAMS-Ensemble daily forecasts ($CAMS_{2019}$) according to Eq. 1:

$$Exp_{2020} = \frac{Obs_{2019}}{CAMS_{2019}} \, CAMS_{2020} \tag{1}$$

2020 CAMS–Ensemble forecasts account for actual meteorological conditions and seasonal changes in emission source strengths, ignoring lockdown measures. The ratio $Obs_{2019}/CAMS_{2019}$ represents the time dependent normalisation of CAMS forecasts to the observations performed at each measurement site, as estimated from 2019 data. Applying this normalisation factor to CAMS 2020 forecasts aims at correcting for the bias between CAMS forecasts and observation data, which can vary across the year. It should be noted that only sites for which forecasts
and observations reasonably agreed ($R^2 \geq 0.5$) across February – May 2019 were considered in this study (see Table S4). Obviously, expected concentrations ($Exp_{2020}$) cannot be compared to observations ($Obs_{2020}$) on a daily basis, since $Exp_{2020}$ values are affected by random variations in the daily $Obs_{2019}/CAMS_{2019}$ ratio. Instead, mean $Obs_{2020}/Exp_{2020}$ ratios were compared for the 3 periods A = 17 February – 8 March 2020 (before lockdowns), D = 23 March – 3 May 2020 (during lockdowns), and P= 11 May – 31 May 2020 (after lockdowns). The statistical
significance of the difference in $Obs_{2020}/Exp_{2020}$ ratios between the 3 time periods A, D, and P was assessed by applying a 2-sided $t$ test to the averages $\bar{A}$, $\bar{P}$, and $\bar{D}$, defined as:

$$\bar{A} = mean\left(log\frac{Obs_{2020}}{Exp_{2020}}\right)_A, \bar{D} = mean\left(log\frac{Obs_{2020}}{Exp_{2020}}\right)_D, \bar{P} = mean\left(log\frac{Obs_{2020}}{Exp_{2020}}\right)_P \tag{2}$$

The null hypotheses ($\bar{D} = \bar{A}$ and $\bar{D} = \bar{P}$) were tested at the 95% confidence assuming unequal variances.

The mean $Obs_{2020}/Exp_{2020}$ ratios plotted and discussed below were calculated as:

$$\langle A \rangle = 10^{\bar{A}}, \langle D \rangle = 10^{\bar{D}}, \text{ and } \langle P \rangle = 10^{\bar{P}} \tag{3}$$

Particle light absorption Ångström exponent (AÅE) values were calculated as the slope of the linear regression between the logarithm of the particle light absorption coefficients and the logarithm of the wavelengths (WL) of the light sources used in the multi-wavelength absorption photometers across the whole WL range available below 900 nm. WL ranges were different across the various sites (Table S3) but constant at each site. AÅE values were
calculated across the ultraviolet – near infrared range (370 – 880 nm) for the urban sites in Lille FR), Bern (CH), Athens (GR), Nicosia (CY) and the regional background sites SIR (FR), PAY (CH), IPR (IT) and MSY (ES). At Brussels (BE), measurements were available at 370 and 660 nm, and at ARN (ES), HYY (FI) and KOS (CZ), in the visible range (470 – 660 nm) only.

The ratio between the number of small particles ($N_{small}$) and the "total" number of particles ($N_{tot}$) was calculated
from PNSD measurements. $N_{small}$ was calculated by integrating PNSD from 15 to 70 nm at all sites, except Copenhagen and RIS (DK), for which PNSD lower bound was 41 nm. $N_{tot}$ was calculated by integrating PNSDs from 15 nm to the upper bound of the distribution at all but both sites in Denmark (41 nm). The upper bound was 800 nm at most but not all sites (Table S3), and was constant at each site across the time period 2017 -2020.



Particle light absorption coefficients and PNSDs are not computed by the CAMS model. Therefore, 2017 – 2019
measurement data were used to calculate the expected values of the Absorption Ångström exponent (AÅE) and
the contribution of small particles to the total particle number concentration ($N_{small}$ / $N_{tot}$) for sites at which
measurements were available for at least 2 years between 2017 and 2019 across the studied period (17 February
– 31 May). Both being intensive variables (i.e. intrinsic aerosol properties), these variables are much less sensitive
to weather conditions than e.g. atmospheric concentrations. Daily values expected for 2020 ($Exp_{2020}$) were
calculated as the average of 2017 – 2019 data, and lockdowns' impacts were assessed comparing the arithmetic
means of $Obs_{2020}/Exp_{2020}$ ($\bar{A}$, $\bar{P}$, and $\bar{D}$) for the 3 time periods A, D, and P, as described above.

**3 Results and discussion**

**3.1 Impact of lockdown measures on road traffic intensity**

Biases in traffic intensity estimates derived from mobility data have been reported in business as usual conditions
(Meppelink et al., 2020). However, data relative to the lockdown period in the USA have highlighted a clear
covariation between Apple® mobility data and gasoline demand (Ou et al., 2020), which is in turn a robust
indicator for the cumulative distance covered by cars. We compared monthly mean motor fuel consumptions and
Apple® driving direction requests for the 12 countries of this study for which data were available. Table S5 shows
that gasoline national consumptions are generally well correlated with country mean driving direction requests
across January – May 2020, while diesel consumptions are anti-correlated or not-significantly correlated with
driving direction requests in all countries but 3 (FR, IT, ES). This suggests that Apple® driving direction requests
are good qualitative proxies for personal car traffic but not for commercial (diesel powered) vehicular traffic. This
is confirmed by data from Athens, for which a reduction between A and D of up to 70% and 40% in Light Duty
Vehicle and Heavy Duty Vehicle traffic, respectively, was reported (Eleftheriadis et al., 2021), to be compared
with a -73% decrease in driving route requests. However, changes in gasoline consumption are everywhere less
than the variations in Apple® driving route requests (range 38% –88%, average 59%) as shown in Table S5.

For the set of cities where urban measurement sites were located, the Apple® mobility data show that driving
route requests dropped by -31 % (Helsinki) to -90 % (Seville) between periods A and D, and increased again by
+40 % (Helsinki) to +270 % (Paris) between periods D and P (Figure 2). The data recorded in the areas
surrounding the regional background sites and/or the urban sites show similar variations, except for the Vysocina
region (central Czech Republic) where driving route request numbers (-27 %) fell much less than in Prague (-60
%), and reached again their pre-lockdown value (period A) when lockdown measures were relaxed (period P).

According to monthly mean data available from CYSTAT, the road transport index in Cyprus decreased by -21
% in March 2020, by further -35 % in April 2020, and increased again by +18 % in May 2020 (Figure S1), in line
with the mobility data collected for the other sites.

These data suggest that at least passenger car traffic strongly decreased as lockdown measures were implemented
(period D) in all countries considered in this study, and particularly in Belgium, France, Italy, Spain, Greece and
Cyprus. They also suggest that the road traffic intensity was largely greater by the end of May 2020 (period P)
than during lockdown periods (period D) at all sites, but without reaching the intensities observed before (period
A) at sites in Belgium, France, Italy, Spain, Greece and possibly Cyprus. However, 2020 monthly automotive fuel
consumption data suggest that light and heavy-duty vehicle traffic (diesel) was much less reduced than private car
(diesel + gasoline) traffic during the first lockdown period across Europe (Section 2.1, Table S5).

**3.2 Impact of lockdown measures on PM mass concentrations**

It is reminded here again that only sites for which the correlation between modelled and measured PM mass
concentrations was satisfactory (positive slope, $R^2 \geq 0.5$) across the February – May 2019 period were considered
(Table S4).

Figure 3 shows the mean observed / expected $PM_{2.5}$ and $PM_{10}$ mass concentration ratios (Eq. 3) at urban sites
during the 3 time periods defined as before (A), during (D) and after (P) the lockdowns. The differences between
these 3 values represent our estimates of the lockdown measures' impacts on pollutant atmospheric
concentrations. As already observed across Europe (EEA, 2020; Shi et al., 2021) and the USA (Bekbulat et al.,
2021), there was no systematic response of PM mass concentrations to the lockdown measures at urban sites.
Indeed, the implementation of lockdown measures in March led to statistically significant decreases in $PM_{2.5}$
concentrations in Oslo, Rotterdam, and Barcelona (3 among 10 sites), and to a significant decrease in $PM_{10}$
concentration in Barcelona and Seville (2 out of 12 sites) only. On average (Table 2), the implementation of the



lockdown measures resulted in minor increases in PM$_{2.5}$ and PM$_{10}$ mass concentration of +1 ± 42 % and +5 ± 33 %, respectively. In contrast, the relaxation of lockdown measures in May led to statistically significant increases in PM$_{2.5}$ concentrations in Helsinki, Rotterdam, Brussels, Prague, Bern and Barcelona (6 out of 10 sites), and to a significant increase in PM$_{10}$ concentrations in Rotterdam, Lille, Prague, Paris, Milan, Barcelona, and Nicosia (7 out of 12 sites). On average, the relaxation of the lockdown measures led to PM$_{2.5}$ and PM$_{10}$ concentration increases of +26 ± 21 % and +26 ± 24 %, respectively. Where significant, lockdown measures had very similar impacts on PM$_{2.5}$ and PM$_{10}$ concentrations. Lockdown impacts on PM concentrations in cities did generally not fully reflect the variations in road traffic intensity expected from the driving road request data (Figure 2).

Advection from surrounding areas have been shown to contribute to PM concentrations in European cities (e.g. Kiesewetter et al., 2015; Thunis et al., 2018; Pommier et al., 2020). For example, modelling indicates that the contribution of sources outside the greater city contribute from 35 % (Paris, Athens) to 94 % (Nicosia) to PM$_{2.5}$ urban background concentrations in the cities considered here (Thunis et al., 2017).

Figure 4 shows the mean observed / expected PM$_{2.5}$ and PM$_{10}$ mass concentration ratios (Eq. 3) during the 3 time periods "before" (A), "during" (D) and "after" (P) the lockdowns at regional background sites in the regions of the cities mentioned above. The implementation of lockdown measures led to statistically significant decreases in PM$_{2.5}$ concentration in BIR (NO), and in PM10 concentrations in BIR and MEL (DE) only (3 out of 16 entries), while their relaxation resulted in significant increases in PM$_{2.5}$ concentrations in BIR, CBW (NL), MEL, KOS (CZ), SIR (FR), and in PM$_{10}$ concentrations in BIR, CBW, MEL, SIR, PAY (CH), and IPR (IT), i.e., 11 out of 16 entries. At regional background sites, lockdown implementations resulted on average, in +2 ± 39 % and +15 ± 42 % increases in PM$_{2.5}$ and PM$_{10}$ mass concentrations, respectively. Their relaxation resulted in further +38 ± 43 % and +28 ± 10 % increases in PM$_{2.5}$ and PM$_{10}$ concentrations, respectively. Comparing PM data with the driving route request data in Figure 1 (bottom) suggests no significant impact of private car traffic intensity on regional background PM levels.

There is generally no correspondence between significant lockdown impacts on PM concentrations at twin sites (urban and regional background sites located in the same area), except for Oslo-BIR (PM decrease from A to D), and Rotterdam-CBW, Prague-KOS, Paris-SIR, and Milan-IPR (PM increase from D to P).

Due to the multiplicity of PM primary and secondary sources, further atmospheric variables such as gaseous pollutant concentrations and PM intrinsic characteristics shall be examined to investigate the lack of dramatic drops in PM mass concentrations resulting from the reduction in private car traffic when lockdown measures were implemented at the sites considered in this study.

## 3.3 Impact of lockdown measures on NO₂ concentrations

Road traffic is the major source of NO$_x$ (nitrogen oxides) in Europe (EEA, 2020). For the 15 cities considered in this study, the contribution of road traffic to annual NO$_x$ emissions ranges from 35 % in Rotterdam to 95 % in Athens (Degraeuwe et al., 2019). Road-traffic intensity variations are therefore expected to significantly affect NO$_2$ concentrations. Diesel engines are by far the largest contributors to road traffic NO$_x$ emissions in most countries across Europe, a noticeable exception being Greece (2016 data). However, for all 13 countries considered, the passenger car fleet emits the largest share of NO$_x$ (60 – 90 %), far ahead that of the light duty plus heavy duty vehicle fleet. Since mobility restrictions presumably affected mostly passenger cars (Section 3.1), dramatic variations in driving route requests (as a proxy for vehicle kilometres) are expected to result in significant changes in road traffic NO$_x$ emissions.

Lockdown measure implementations led to statistically significant decreases in NO$_2$ concentrations in Helsinki, Copenhagen, Rotterdam, Brussels, Lille, Paris, Bern, Milan, and Barcelona, i.e., in 9 among 13 cities. At sites where no significant reduction in NO$_2$ concentrations occurred, there was no significant decreases in nitrogen oxide (NO) concentrations either (Figure S2), indicating no substantial abatement of NO$_x$ emissions. On average, the implementation of lockdown measures resulted in NO$_2$ concentration decreasing by -29 ± 17 %. Lockdown measure relaxations led to significant rebounds in NO$_2$ concentrations in Copenhagen, Brussels, Lille, Prague, Paris, Milan, and Seville (7 among 13 cities). The mean increase in NO$_2$ concentration resulting from the lockdown termination was +31 ± 30 %. Although the impact of the lockdown measures was more systematic for NO$_2$ than for PM concentrations (Figure S3), there is no significant correlation between the impact on NO$_2$ concentration and the reduction in driving route requests from periods A to D, and only a marginally significant



correlation between the impact on NO$_2$ concentration and the increase in driving route requests from periods D to P.

Lockdown measures also resulted in significant decreases in NO$_2$ concentrations at 7 of the 12 regional background sites (HYY, FI; CBW, NL; SIR, FR; PAY, CH; IPR, IT; CYP, CY). Their relaxation led to significant increases in NO$_2$ at 3 sites only, namely CBW (NL), MEL (DE), and SIR (FR). On average, the implementation and relaxation of lockdown measures resulted in a -17 ± 24 % decrease and +27 ± 50 % increase in NO$_2$ concentration at regional background sites, respectively. There is no statistically significant correlation (95 % confidence level) between the lockdown impact on NO$_2$ concentrations and the changes in route request data from periods A to D and D to P at the regional background sites.

Figure 5 shows that there is generally no matching in NO$_2$ ratio variations from periods A to D, and from periods D to P, between urban sites and regional background sites in surrounding areas (with a few exceptions including Paris-SIR and Bern-PAY), which suggests that NO$_2$ concentrations at urban and regional background sites are controlled by different sources and /or atmospheric processes.

The lack of systematic correlation between the variations in the Apple® driving route request index and the changes in NO$_2$ concentrations due to the lockdown implementation and relaxation measures suggests that the linkage between passenger car traffic and NO$_x$ emissions was not that straightforward under such circumstances. However, NO$_2$ being an important precursor of secondary PM, and the vehicles that emit most NO$_x$ being also those which emit most primary PM, the lack of dramatic impact of lockdowns on PM concentrations compared to their clear effect on NO$_2$ concentrations at many sites emphasises the variety of PM sources and the complexity of secondary formation processes.

### 3.4 Impact of lockdown measures on O$_3$ concentrations

Implementations of lockdown measures induced statistically significant increases in O$_3$ concentrations in Brussels, Lille, Paris, Barcelona, and Nicosia, i.e., in 5 cities out of 12, Milan being on the edge (Figure 6, top). There were no cities where lockdown measures led to a significant decrease in O$_3$. This is consistent with photochemical O$_3$ production not being limited by the availability of NO$_x$ in urban areas, and with a reduction of O$_3$ titration by NO as resulting from an abatement in NO$_x$ emissions during the lockdown periods. On average, the implementation of the lockdown measures resulted in an increase of + 11 ± 23 % in O$_3$ concentration in cities. Also at the regional background sites BIR (NO), KOS (CZ), SIR (FR), IPR (IT) and ARN (ES), the effect of the lockdown measures was a significant increase in O$_3$ concentrations, and at no sites was a significant decrease in O$_3$ detected. This is again consistent with an excess of NO$_x$ in O$_3$ photochemical formation at those sites, at least during this period of the year (February – April). The significant increase in O$_3$ after the lockdown measure relaxation (period P) in BIR (NO), and ARN (ES) could be explained by a shift in the O$_3$ photochemical production to the "NO$_x$ limited" regime at these regional background sites, resulting from increased emissions of biogenic volatile organic species in May. However, the mean impact of the lockdown measures implementation and relaxation on O$_3$ concentration at regional background sites was quite marginal (+17 ± 24 % and +4 ± 5 %, respectively).

Increased O$_3$ concentrations reflect an increase in the oxidizing capacity of the atmosphere. The increased oxidizing capacity of the atmosphere was invoked to explain the lack of systematic decrease in PM concentrations resulting from the lockdown measures (e.g. Kroll et al., 2020): the expected decrease in PM primary emissions would be compensated (or even over-compensated) by an increased production of secondary aerosol resulting from a faster oxidation of PM gaseous precursors to condensable material. Actually, increased aerosol surface area and Aitken mode particle growth rates were observed for the lockdown period in Athens (Eleftheriadis et al., 2021), together with increases in O$_3$ and PM$_{2.5}$ concentrations. This hypothesis is to some extent also supported by recent modelling works (Clappier et al., 2021) suggesting that in the areas surrounding Rotterdam, Bern, Milan, and Barcelona, reduction in NO$_x$ emissions would lead to enhanced secondary PM formation resulting from the increased oxidizing capacity of the atmosphere. However, the magnitude of this phenomenon during the 2020 lockdown could only be assessed on the basis of PM chemical composition data, which were not collected under this study.

In the following sections, other possibilities will be examined by making use of specific aerosol properties, which are not part of the air pollution regulated metrics.

### 3.5 Impact of the lockdown measures on aerosol intrinsic characteristics



A reason why lockdown impacts on PM mass concentrations were smaller than expected could be the compensation of the decrease in road traffic emissions by the increase in domestic heating emissions, resulting from people "staying-at-home" (Altuwayjiri, et al., 2021). Since wood (or wood pellets) is one of the fuels used for domestic heating, any decrease in road traffic compared to domestic heating emissions would result in an

increase of the AÅE. We also deemed it important to assess how clear was the lockdown effects on primary particle emissions from vehicle engines. This is why the contribution of small particles ($D_p$<70 nm) to the total particle number ($N_{small}$ / $N_{tot}$) was examined. The measurement data needed to calculate these intensive variables were not available for all the 28 sites considered in this study (see Table S3). Therefore, data from urban and regional background sites are not split in separate figures for these two variables.

**3.5.1 Particle light absorption spectral dependence**

A statistically significant impact of the lockdown measure implementation and relaxation on the particle light absorption spectral dependence was detected in Lille (FR), Athens (GR) and ARN only (Figure 7). At both urban sites, 2020 AÅE values were very similar to the 2017 -2019 averages for periods A and P, and significantly greater during the lockdown period (D). AÅE values also significantly increased as lockdown measures were

implemented in Oslo (NO) and SIR (FR), while AÅE values significantly decreased as lockdown measures were relaxed in IPR (IT). No significant change in 2020 AÅE values (as compared with 2017 - 2019) could be observed for the lockdown period (D) in BIR (NO), Brussels (BE), KOS (CZ), Bern (CH), PAY (CH), and MSY (ES). On average (all sites), the AÅE increased by +3 ± 6 % and decreased by -8 ± 28 % as lockdown measures were implemented and relaxed. In short, the expected increase in AÅE resulting from a decrease in particle emissions

from traffic and a stagnation or increase in particulate emissions from wood burning during the lockdown period was not systematically observed across the sites considered in this study. Therefore, increased emissions from domestic heating during the lockdown can have contributed to maintain unexpectedly high PM mass concentrations in certain places across Europe, but this phenomenon was apparently not relevant in many areas.

**3.5.2 Particle number size distribution**

At 3 of the 5 urban sites for which data were available (Leipzig, Athens, and Granada), the implementation of the lockdown measures in 2020 coincided with a statistically significant decrease in the $N_{small}$ / $N_{tot}$ ratio as compared to the same time periods in 2017 – 2019 (Figure 8). A significant increase in this ratio occurred as lockdown measures were relaxed at only 2 amongst the 6 urban sites with relevant data (Copenhagen and Barcelona). On average, the implementation and relaxation of lockdown measures corresponded to a decrease by -7 ± 5 % and an

increase by +6 ± 2 %, respectively, in the $N_{small}$ / $N_{tot}$ ratio.

These observations suggest that the decrease in traffic resulting from the implementation of the lockdown measures led on average to a significant but moderate decrease in the number concentration of primary ultrafine particles (which dominate the 15 – 70 nm size range in urban environments). The lack of complete return to usual PNSDs after the lockdown period ended can be explained by only partial recovery in human mobility in Athens

(GR) and Granada (ES), but not in Leipzig (DE) where mobility almost completely (95 %) recovered.

During the lockdown period (D), statistically significant changes in the contribution of small particles to the whole PNSD ($N_{small}$ / $N_{tot}$) were also detected at 4 out of 6 regional background sites, i.e. BIR (NO), RIS (DK), MEL (DE) and IPR (IT). Variations in RIS and MEL reflected quite well the variations in the nearby cities of Copenhagen and Leipzig. Clear decreases and increases in $N_{small}$ / $N_{tot}$ corresponding to the lockdown measures

implementation and relaxation, respectively, can be noticed at both BIR (NO) and IPR (IT). The variations observed in IPR can easily be related to the variations in the driving route request index for the densely populated and traffic impacted Lombardy region. In contrast, it is surprising to observe such significant changes in the PNSD in BIR, located in a region (Agder) where the driving mobility index remained relatively high, even during the lockdown period. Lockdown measures also had a huge impact on PM mass concentration in BIR (Figure 3), but

providing a specific explanation for the case of BIR is beyond the scope of this study. On average, the implementation and relaxation of lockdown measures coincided with a decrease by -9 ± 13 % and an increase by +11 ± 12 %, respectively, in the $N_{small}$ / $N_{tot}$ ratio at regional background sites.

In short, except for the two sites located in Finland, the lockdown periods coincided with unusual low shares of small particles ($N_{small}$ / $N_{tot}$) at all sites, although differences were not all statistically significant. This is a clear

suggestion that the lockdown measures did have an impact on primary particle emissions. However, considering the huge changes in the driving route request index for a vast majority of sites in this study, the impact on PNSD



was not quite dramatic. This suggests that private cars do not significantly contribute to the overall emission of 15-70 nm particles at the sites we studied, or that the decrease in this specific source was compensated by increases in other sources during the lockdown periods.

## 4 Conclusions

Specific impacts on air pollution of the implementation and relaxation of lockdown measures to prevent the spread of COVID-19 were determined by comparing measurement data with expected data for the period 17 February – 31 May 2020 (Table 2).

Driving direction request data suggest that the level of reduction in car passenger traffic resulting from the lockdown measures was much deeper in the south than in the north of Europe. Our study shows that quite independently from the changes in these human mobility indices, the implementation of the lockdown measures did not result in statistically significant decreases in $PM_{2.5}$ and $PM_{10}$ mass concentrations at most of the European urban sites we considered. On average, lockdown entries into force in March 2020 did not lead to a decrease in $PM_{2.5}$ and /or $PM_{10}$ mass concentrations, with great differences across the various sites though (Figure S3). However, the relaxation of the lockdown measures in May 2020 led to an increase of $PM_{2.5}$ and /or $PM_{10}$ concentrations at more than half of our test cities, resulting in a mean increase of + 26 % in both $PM_{2.5}$ and $PM_{10}$ concentrations. At regional background sites, a significant effect of the lockdown measure implementation was detected at even fewer sites, while their relaxation led to $PM_{2.5}$ and/or $PM_{10}$ mass concentration increases at most of them. The asymmetrical response of PM mass concentrations to the implementation and relaxation of lockdown measures suggests a more complex than expected relationship between road traffic intensity and PM mass concentrations. Considering a series of other atmospheric variables allowed us to partially understand this phenomenon.

$NO_2$ concentrations significantly decreased at 3/5 of the urban sites, and significantly re-increased at 3/10 of them due to the lockdown measure implementation and relaxation, respectively. On average, lockdowns' implementation resulted in -29 % decrease in $NO_2$ concentration, and their relaxation led to a + 31 % rebound. This suggests that mobility restrictions did translate into decreases in road traffic $NO_x$ emissions. However, the magnitude of the changes in $NO_2$ concentrations did not correlate well with the changes in human mobility. This could be due to the fact that driving route request indices differently reflected the number of km driven in various countries, and by differences in the fraction of vehicles complying with different EURO emission standards across Europe.

Lockdown-induced changes in the oxidizing capacity of the atmosphere could have led to the formation of more secondary aerosols during lockdown periods, even from less gaseous precursors. Indeed, we found that $O_3$ concentrations (an indicator of the oxidizing capacity of the atmosphere) significantly increased at 1/2 of both the urban and regional background sites due to lockdown measure implementations. The production of secondary particulate matter could have been boosted at these sites. Decreases in $NO_x$ emissions leading to increased secondary PM concentrations in parts of Europe were actually shown by Clappier et al. (2021) in their modelling work. However, further data regarding the aerosol chemical composition would be needed to determine whether this process significantly affected PM mass concentrations as lockdown measures were implemented at the sites we studied.

Significantly greater AÅE values observed during the lockdown periods at a few sites in Norway, France, Italy, Greece and Spain suggest a relative increase in black carbon emission from wood burning as compared to fossil fuel burning during those periods. Therefore, the decrease in traffic-related PM concentrations could have been at least partially compensated by an increase in domestic heating-related PM concentrations at these sites. However, this phenomenon was apparently not generalised across Europe.

Otherwise, a statistically significant effect on PNSD was observed at most of the sites studied. Moderate -7 % and -9 % decreases in the contribution of small particles to the total particle number concentration occurred on average across the urban and regional background sites, respectively. This indicates that lockdown measures led to a decrease in primary particle emissions (mainly in the 15 – 70 nm range) compared to the production of secondary particles (mainly in the range 100 – a few hundreds of nm). This suggests that measures reducing passenger car traffic (as lockdown measures did) would probably have a larger impact on particle number concentrations (strongly dependent on the number of small particles) than on PM mass concentrations (much more sensitive to the number of particles in the range 70 – several hundreds of nm) in cities.



Our results are in line with previous studies showing the unexpectedly small impacts of lockdowns on PM mass concentrations over Europe and the USA (Bekbulat et al., 2021; Shi et al., 2021; Querol et al., 2021), and are contrasting other findings, which indicated great reductions in PM mass concentrations in several big European and American cities due to the COVID-19 lockdown measures (Chauhan and Singh, 2020; Beloconi et al., 2021; Jiang et al., 2021).

This study contributes to a better understanding of the effect of human mobility constraints on particulate air pollution based on the exceptional circumstances of the COVID-19 lockdowns. We highlighted the complexity of PM mass concentration responses to the COVID lockdown measures implemented across Europe, resulting from a combination of several factors including different levels of stringency in the various European countries, proven compensation between road traffic and domestic heating emissions at some sites, and possible increased formation of secondary PM at other sites. The different contributions of these phenomena at various places across Europe still needs to be quantitatively determined. Nevertheless, the COVID-19 lockdown "experiment" suggests that the on-going decrease in exhaust emissions by the passenger car fleet might have quite contrasting impacts on air quality in European cities.

**Disclaimer**

The information and views set out are those of the authors and do not necessarily reflect the official opinion of the European Commission**.**

**Competing interests**

One of the co-authors is a member of the editorial board of Atmospheric Chemistry and Physics. The peer-review process was guided by an independent editor, and the authors have also no other competing interests to declare.

**Authorcontributions**

Conceptualization and methodology: JPP and EP. Formal analysis: JPP. Investigation: JPP, AM, CH, JS, MP, JO, SM, KW, MM, LP, DvP, AM, CN, CR, NP, SC, MS, JAA, TP, KL, JN, VR, JFdB, AC, OF, JEP, VG, MIG, SV, ED, HDvdG, KEY, WA. Original draft preparation: JPP. Review and editing: EP, AM, CH, MP, JO, SM, AW, LP, HH, AM, AA, NP, SC, MS, JAA, KL, JN, VR, JFdB, AC, KE, HDvdG, WA.

**Data availability**

Observation data from ACTRIS observatories are available at actris.nilu.no and ebas.nilu.no. Observation data from urban sites are available from local Air Quality Monitoring Networks and from www.eea.europa.eu/data-and-maps/data/aqereporting-9. Model forecasts for all sites are available at ads.atmosphere.copernicus.eu/cdsapp#!/dataset/cams-europe-air-quality-forecasts?tab=form.

**Acknowledgements**

The research presented here relied on the data and products made available in open access, by the Copernicus Atmosphere Monitoring Service (CAMS) of the Copernicus Programme of the European Union, and in particular the CAMS Policy Service, //policy.atmosphere.copernicus.eu/.

PNSD and particle light absorption data were provided by the ACTRIS Data Centre developed under the European Union's Horizon 2020 research and innovation programme under grant agreement No 654109 (ACTRIS-2). Data quality assurance has been supported by the ACTRIS-IMP project of the European Commission under grant agreement No 871115.

The Belgian Interregional Environment Agency (Ircel-Celine) is acknowledged for the provision of $PM_{10}$, $PM_{2.5}$, NO, $NO_2$ and $O_3$ data from Brussels.

Czech Hydrometeorological Institute provided air quality monitoring data from Prague (Libus). Conditions of data utilization (in Czech): www.chmi.cz/files/portal/docs/uoco/historicka_data/OpenIsko_data.

Data from Leipzig-West was kindly provided by the Saxon State Office for the Environment, Agriculture and Geology (LfULG).

IMT Nord Europe acknowledges financial support from the Labex CaPPA project, which is funded by the French National Research Agency (ANR) through the PIA (Programme d'Investissement d'Avenir) under contract ANR-



11-LABX-0005-01, and the CLIMIBIO project, both financed by the Regional Council "Hauts-de-France" and the European Regional Development Fund (ERDF). IMT Nord Europe participated in the COST COLOSSAL Action CA16109. The ATOLL site (Lille) is one of the French ACTRIS National Facilities and contributes to the CARA program of the LCSQA funded by the French Ministry of Environment.

Airparif, the agency for air quality monitoring in the Ile-de France region, kindly provided open access to air
quality data.

ARPA Lombardia kindly provided air quality data from Milan.

The Rotterdam EPA (DCMR) kindly provided air quality measurements in Rotterdam.

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



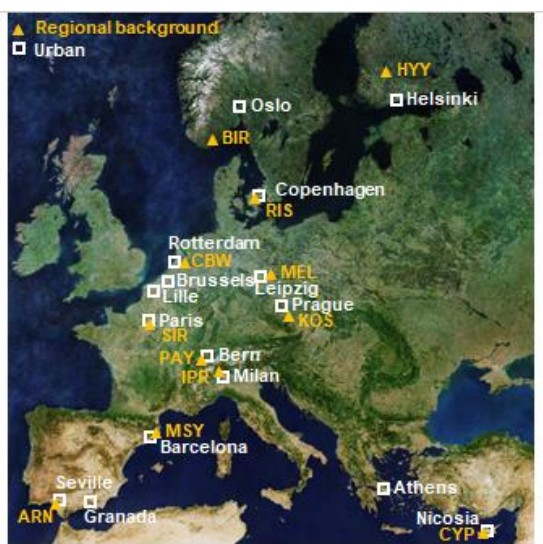


Figure 1. Location of the 28 sites across Europe (map background from ESA).



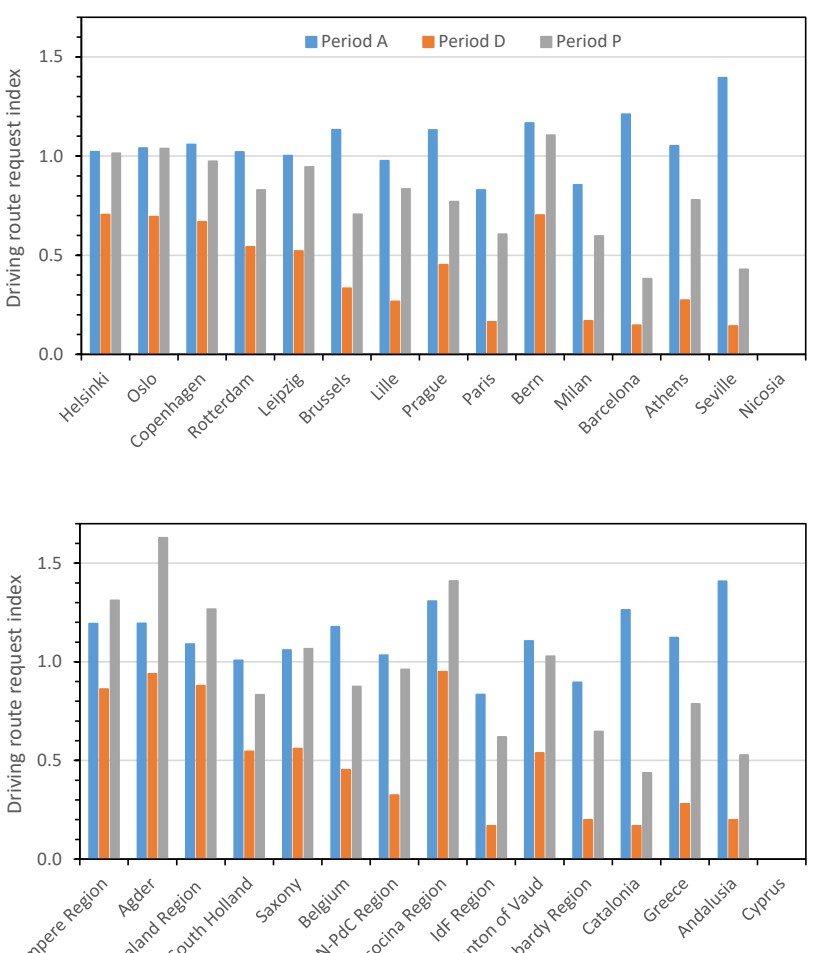

Figure 2. Driving route request indices during period A (17 February – 8 March), D (23 March– 3 May) and P (11 – 31 May) in cities where measurement sites were located (top), and in areas where regional background sites and/or urban sites were located (bottom). Apple® mobility indices are relative to data of 13 January 2020. Data from 11 and 12 May 2020 are missing. N-PdC and IdF stand for the French "Nord-Pas de Calais" and "Ile de France" regions, respectively. No data for Cyprus (see Fig. S1).





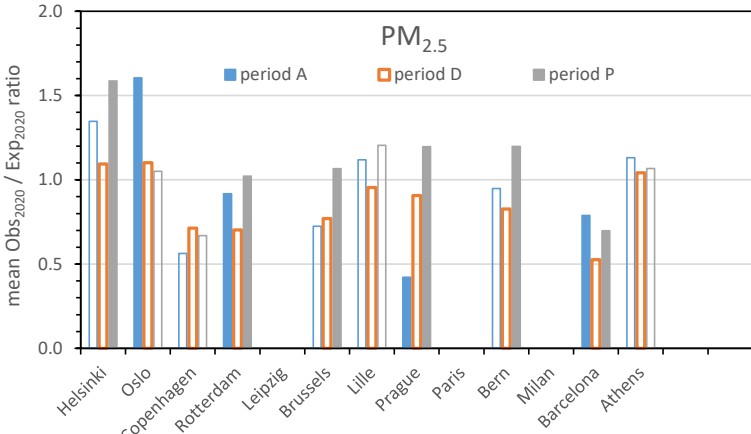


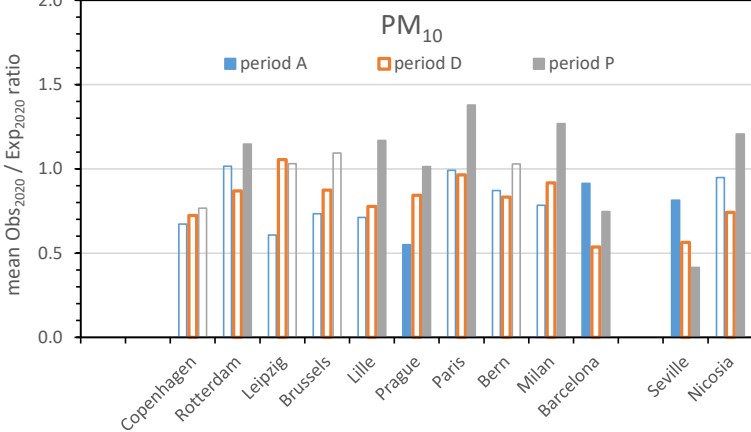

Figure 3. Mean observed / expected $PM_{2.5}$ (top) and $PM_{10}$ (bottom) concentration ratios (Eq. 3) during periods A (before), D (during lockdowns) and P (after) at urban sites. Filled bars indicate means which are statistically different from the mean during lockdown periods (D).


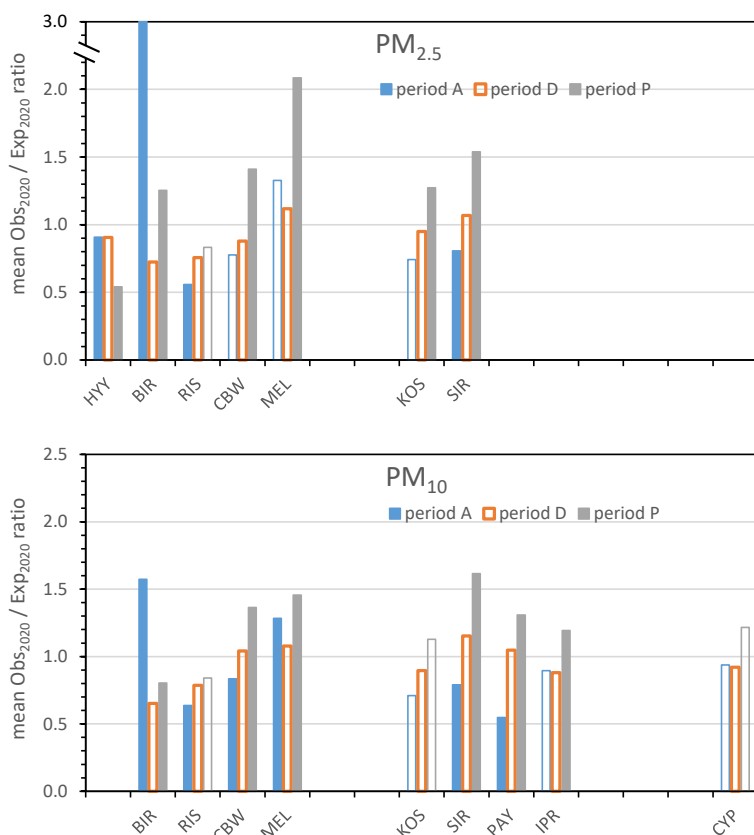

Figure 4. Mean observed / expected $PM_{2.5}$ (top) and $PM_{10}$ (bottom) concentration ratios (Eq. 3) during periods A (before), D (during lockdowns) and P (after) at regional background sites. Filled bars indicate means which are statistically different from the mean during lockdown periods (D).




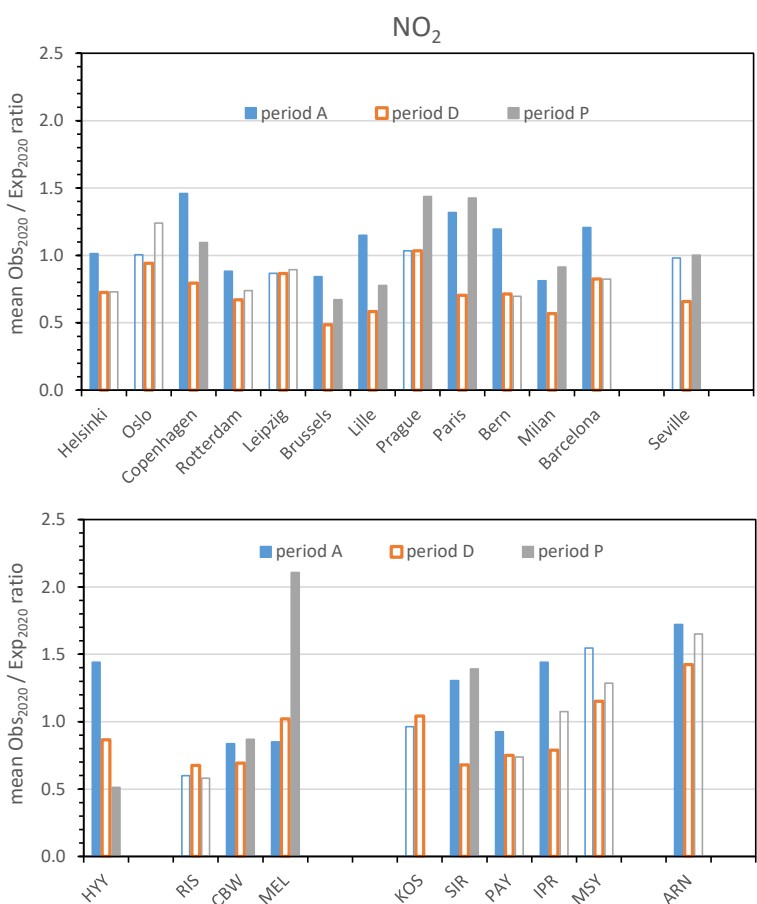

Figure 5. Mean observed / expected NO₂ concentration ratios (Eq. 3) during periods A, D and P. Filled bars indicate means which are statistically different from the mean during lockdown periods (D). No data available from KOS for time period P.



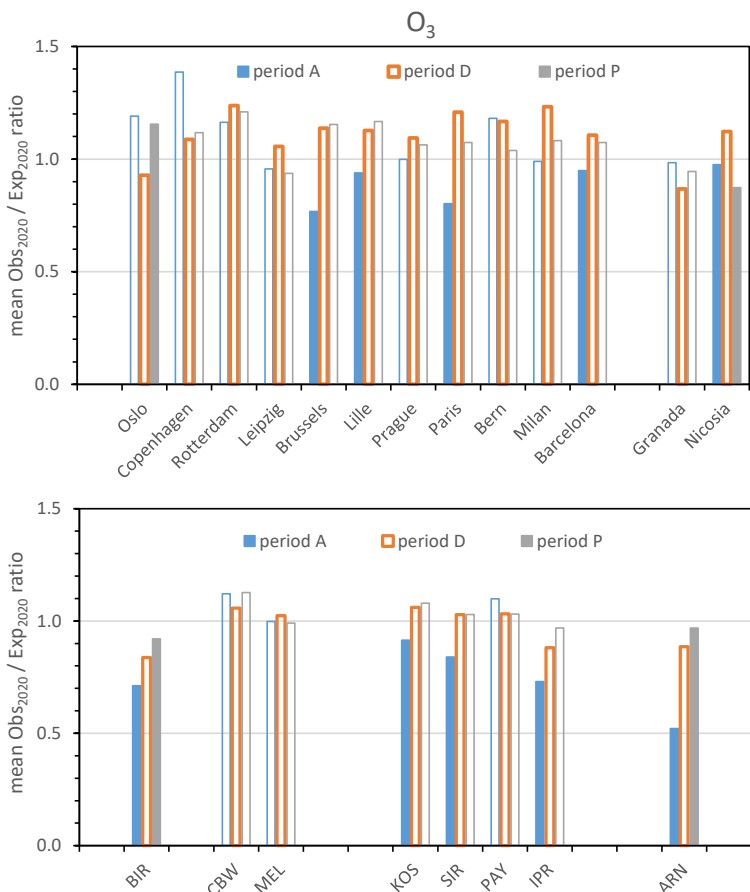

Figure 6. Mean observed / expected $O_3$ concentration ratios (Eq. 3) during periods A, D and P. Filled bars indicate means which are statistically different from the means during lockdown periods (D).



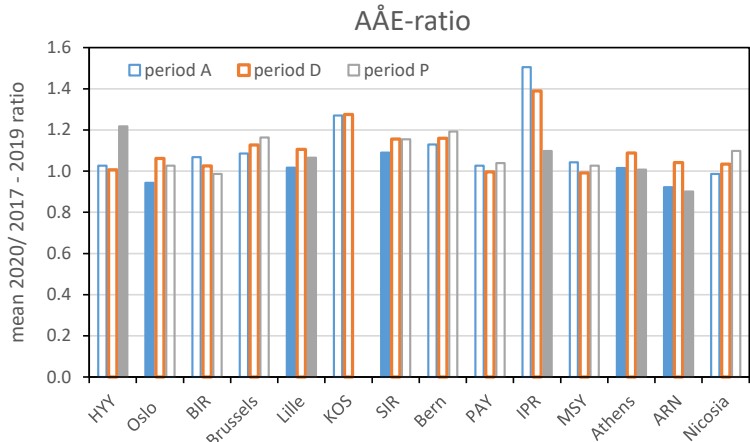

Figure 7. Mean 2020 / <2017 – 2019> AÅE ratios during periods A, D and P. Filled bars indicate means which are statistically different from the means during lockdown periods (D). No data available for KOS during period P.

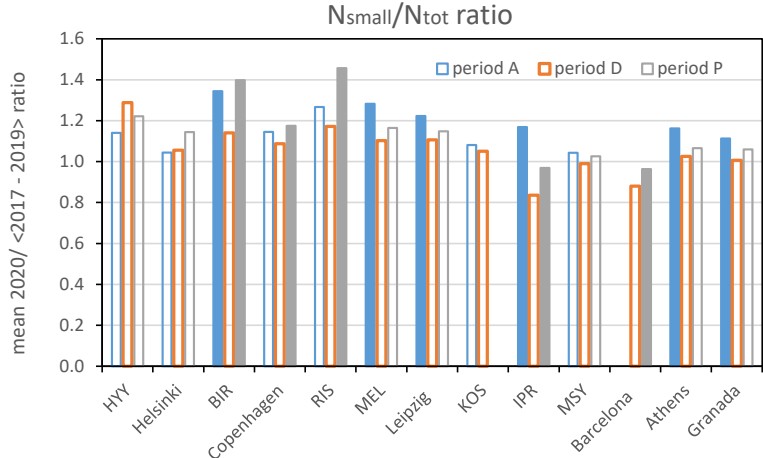

Figure 8. Mean 2020 / <2017 – 2019> $N_{small}$ / $N_{tot}$ ratios during periods A, D and P. Filled bars indicate means which are statistically different from the means during lockdown periods (D). No data available for KOS during period P, and for Barcelona during period A.



Table 1. Measurement site details.

| | Urban sites | | | | Regional background sites | | | |
|---|---|---|---|---|---|---|---|---|
| Country | Site | Type | Latitude (°N) | Longitude (°E) | Site | Code | Latitude (°N) | Longitude (°E) |
| FI | Helsinki[1] | Background | 60.19 | 24.95 | Hyytiala | HYY | 61.85 | 24.28 |
| NO | Oslo | Background | 59.92 | 10.77 | Birkenes | BIR | 58.39 | 8.25 |
| DK | Copenhagen | Traffic | 55.67 | 12.57 | Risoe | RIS | 55.64 | 12.09 |
| NL | Rotterdam | Background | 51.93 | 4.23 | Cabauw | CBW | 51.97 | 4.92 |
| DE | Leipzig[2] | Background | 51.32 | 12.30 | Melpitz | MEL | 51.53 | 12.93 |
| BE | Brussels | Background | 50.80 | 4.36 | | | | |
| FR | Lille | Background | 50.63 | 3.09 | | | | |
| CZ | Prague | Background | 50.01 | 14.45 | Kosetice | KOS | 49.57 | 15.08 |
| FR | Paris | Background | 48.89 | 2.35 | SIRTA | SIR | 48.71 | 2.16 |
| CH | Bern | Traffic | 46.95 | 7.44 | Payerne | PAY | 46.81 | 6.95 |
| IT | Milan | Background | 45.48 | 9.23 | Ispra | IPR | 45.82 | 8.64 |
| ES | Barcelona | Background | 41.39 | 2.12 | Montseny | MSY | 41.77 | 2.35 |
| GR | Athens | Background | 38.00 | 23.82 | | | | |
| ES | Seville | Background | 37.35 | -6.06 | El Arenosillo | ARN | 37.10 | -6.73 |
| ES | Granada | Background | 37.16 | -3.61 | | | | |
| CY | Nicosia | Background | 35.14 | 33.31 | Agia Marina | CYP | 35.04 | 33.06 |

(1) Helsinki PNSD data are from the University of Helsinki science campus area located at 60.20 °N, 24.96 °E.
(2) Leipzig PNSD data are from the Leipzig Science Park area located at 51.35°N, 12.43°E.


Table 2. Mean impacts of the lockdown measures implementation and relaxation (%).

| Variable | Site Type | Impact of lockdown measures | | | |
|---|---|---|---|---|---|
| | | implementation | | relaxation | |
| $PM_{2.5}$ | Urban | +1 % | ± 46 % | +26 % | ± 22 % |
| | Regional background | +2 % | ± 39 % | +38 % | ± 43 % |
| $PM_{10}$ | Urban | +5 % | ± 33 % | +26 % | ± 24 % |
| | Regional background | +15 % | ± 42 % | +28 % | ± 10 % |
| $NO_2$ | Urban | -29 % | ± 17 % | +31 % | ± 30 % |
| | Regional background | -17 % | ± 24 % | +27 % | ± 50 % |
| $O_3$ | Urban | +11 % | ± 23 % | -3 % | ± 12 % |
| | Regional background | +17 % | ± 24 % | +4 % | ± 5 % |
| AÅE | Urban | +7 % | ± 4 % | 0 % | ± 5 % |
| | Regional background | 0 % | ± 7 % | -14 % | ± 37 % |
| $N_{small}/N_{tot}$ | Urban | -7 % | ± 5 % | +6 % | ± 2 % |
| | Regional background | -9 % | ± 13 % | +11 % | ± 12 % |