# Peer review of "Impact of 2020 COVID-19 lockdowns on particulate air pollution across Europe"

_EGUsphere, 2023_

## Referee Comment (RC1)

**Summary:**

This research mainly studies how the 2020 pandemic-resulted lockdown affects the concentrations of atmospheric pollutants in the ambient environment. It utilizes observational data, modeling results, and related statistical analyzes of particulate matters (PM) and PM's associated parameters, nitrogen oxides ($NO_x$), and ozone ($O_3$) at the ground level to illustrate the correlations between the changes of pollutants' concentration and different lockdown phases. The observational data in each region contains at least one urban area site and one regional background site. By doing this, the authors revealed the relative contributions of the anthropogenic factors in the total emission change. The correlation analysis shows the reduction of activities in transportation sectors during the lockdown doesn't necessarily lead to a decrease in PM in the studied European cities. The increasing particle absorption angstrom exponent (AAE) suggests an enhanced emission of biomass burning from the domestic heating sources during the lockdown period. And the observational evidence of increased atmospheric oxidation states facilitates the formation of secondary aerosols. Therefore, the reduction of primary emissions from vehicles was possibly compensated by both the increase of domestic emissions and the new particle formations. It needs more observational constraints to quantify the complex mechanism related to the change of PM concentrations.

**General suggestions:**

1. Three lockdown phases were defined in this study. I suggest that the authors would add a figure to illustrate what principle/index you used (i.e., the proxy mobility data or any other ancillary index) to divide the three periods (e.g., an index which shows an obvious change at the beginning and the ending of the "during lockdown" phase).
2. PNSD and AAE coefficients are computed based on 2017-2019 historical data. Did they follow the same estimation process as it shows in Eqn 1-3? Then why are the other pollutants data (PM, NOx, etc.) based only on the 2019 data? A longer-period base may be better to exclude the influences of severe weather climatology in certain years.
3. Histogram figures: I suggested that showing all the studied measurement cites name even if certain data in some cities/areas may be unavailable, and in a fixed order as well. For example, in Fig. 2A, there are 15 cities (out of 16) shown in the x-axis (Granada is missing). In Fig 6A., Helsinki and Athens data are missing, Seville is substituted by Granada.
4. The main results are shown in the format of histogram plots and tabulated statistics. I suggest that the authors diversify the illustration/visualization style of the outcomes of this study. Some main statistical conclusions of this study are not necessarily drawn from the comparisons between mobility index and pollutants concentration at each location. Such collective significance (or insignificance) can be illustrated in correlation scatter plots (with confidence intervals).
5. Table S3 shows the instruments and the related wavelength range for AAE measurements. Here, according to the definition of AAE, what exact wavelength pairs did the authors use to compute AAE value? Did you use (1) the lower-upper-bound wavelength pair or (2) a fixed

wavelength pair which all types of instruments cover? (The later one makes more sense to me)

6. The variation of AAE value doesn't necessarily mean the change of BC amount in the total particles. Biomass burning contributes a lot of light-absorbing organic carbon (i.e., brown carbon) particles which have a distinct absorption spectrum than black carbon. A detailed analysis of chemical composition of light-absorbing particles may exceed the scope of this study, but it is worth mentioning the contribution of other species which contribute a different AAE value.

**Line-by-line suggestions:**

1. Line-187: Is $N_{small}/N_{tot}$ here is defined as $N_{15-70nm}/N_{15-800nm}$? Did the authors use the same 800 nm upper bound for those sites which have a larger (or lower) upper bound for PNSD? Using different size limits to measure the small particles fraction may lead to biases for a group statistical analysis.

2. Line-193: "these variables are much less sensitive to weather conditions than e.g. atmospheric concentrations." I think it is not strictly correct here to state AAE and PNSD are less sensitive to weather conditions, given that the weather or meteorological conditions play important roles in particle transport and formation.

3. Line-318: "at no sites", there indeed a few sites show $O_3$ decrease in Fig. 6.

4. Table 1: The first 16 sites are "urban sites" and the others are associated "regional background" sites according to the main text. What does the "background" in column "Type" of the first 16 sites stand for? Please clarify if "background" here stands for any specific measurement or meteorological conditions.

5. Table 2: How did the authors compute the percentage of increase or decrease in this table? Are they percentage change of the ratio $Obs_{2020}/Exp_{2020}$? If so, what is the base value for each ratio (Is the base value the historical ratio magnitude, or the ratio of the previous lockdown phase in 2020)?

---

## Author Comment (AC1)

We would like to thank both referees for their comments and suggestions regarding our manuscript ACP-2023-434, which were all carefully considered and addressed as described below.

Referee #1

General suggestions:

1-    Apple® driving direction request index time series have been included as an additional piece of information in the Supplement as suggested (Figure S1).

2-    As described in Section 2.5, all pollutant concentrations observed in 2020 are compared to expected values computed from 2020 model data, and 2019 measurement and model data. However, the particle number size distributions and aerosol light absorption spectra are not computed by the model. Therefore 2020 small particle number ratio ($N_{small}/N_{tot}$) and the Absorption Ångström Exponent (AÅE) are compared to 2017-2019 averages. Since both are intensive variables, they are insensitive to the meteorological conditions that control air pollution dispersion. Considering that black carbon and primary particle emissions may have changed across the last decade (due to e.g., the evolution of the vehicle fleet and domestic heating technologies), we deemed reasonable to consider data from 2017 – 2020 rather than from a longer period of time.

3-    Data from the various sites are constantly shown according to the same sequence (from North to South) in all Figures from Figure 2 to Figure 6. We think that including the names of the sites for which the lockdown effect cannot be assessed (because data are lacking or not meeting our selection criteria) would not bring any additional information. For the same reason, the lockdown effect could be assessed for different variables in Seville and Granada. These cities sitting in the same region of Spain, at about 10' difference in latitude (< 20 km), the choice was made to swap from one to the other to keep the one-to-one correspondence between cities and regions throughout the manuscript.

4-    As suggested, scatter plots showing regressions between the changes in atmospheric pollution and the changes in human mobility index have been included in the Supplement to illustrate the lack of significant correlation between these variables, both at urban and regional background sites (Figures S3 and S4).

5-    As described in Section 2.5 "Particle light absorption Ångström exponent (AÅE) values were calculated as the slope of the linear regression between the logarithm of the particle light absorption coefficients and the logarithm of the wavelengths (WL) of the light sources used in the multi-wavelength absorption photometers across the whole WL range available below 900 nm. WL ranges were different across the various sites (Table S3) but constant at each site.". We aimed at determining the sites at which the lockdown period corresponded to a significant change in the AÅE, beyond the seasonal variation observed across 2017-2019. Therefore, the comparability of the AÅE values across the various sites is not as essential as the consistency of the method for determining the AÅE at each site across 2017-2020.

6-    In line with Referee #1's comment, the variations in the AÅE value were used as an indicator of the change in the aerosol light absorption spectrum. Such changes have traditionally been attributed to variations in the contribution of Brown Carbon to light absorption, which was shown to be linked to the relative contributions of wood and fossil fuel burning emissions to light absorbing aerosol concentrations (e.g., doi.org/10.1016/j.scitotenv.2022.160434). Other species which contribute to higher AÅE values include desert dust (e.g., doi.org/10.1016/j.atmosres.2014.10.015) but no such event could be detected that would significantly affect the mean 2020 / 2017-2019 ratio.

Line-by-line suggestions:

1-    Lines 187-188 read "The upper bound was 800 nm at most but not all sites (Table S3), ….". Table S3 shows indeed that the PNSD upper bound was different from 800 nm (range 478 – 1000 nm) at 6 sites. However, the number of particles in this size range is always negligible compared to the

number of particles in the range 15 – 500 nm, as illustrated in the example of a desert dust outbreak (doi: 10.5194/acp-16-1081-2016). As for the AÅE, we did not compare the $N_{small}$ / $N_{tot}$ ratio amongst the various sites, but looked for changes in this ratio during the lockdown period for each site, taking into account seasonal variations observed across 2017 – 2019. As stated in Line 188, "The upper bound … was constant at each site across the time period 2017 -2020.", which made this comparison possible.

2-    Line 193: Both the AÅE and the $N_{small}$ / $N_{tot}$ variables measured at a given site are indeed depending on weather conditions due to e.g., pollution transport patterns and secondary aerosol formation, but they are much less sensitive than extensive aerosol variables, which are strongly influenced by the horizontal and above all vertical dispersion of pollutants.

3-    Line 318: Figure 6B shows that the increases in $O_3$ resulting from the lockdown implementation were not statistically significant.

4-    Table 1: The revised version of the manuscript includes a footnote to Table 1, which specifies that: "Background" and "Traffic" stand for "urban background" and "traffic" sites, respectively.

5-    Table 2: In the revised version of the manuscript, Table 2 caption specifies that the mean impacts of the lockdown measures implementation and relaxation (%) are computed from the 2020 observed / expected ratios shown in Figures 2 to 8. Equations 4a and 4b used to calculate these mean impacts have been introduced in the revised version of the manuscript.

Referee #2

General comment:

As suggested, we have better specified the scientific objectives of our study in the introduction and expanded our discussion on the significance of our findings in the conclusion. A major finding of our study is that there is no simple explanation for the unexpectedly low impact of lockdowns on particulate air pollution that applies to all the sites we considered throughout Europe. To investigate possible changes in the rate of formation of secondary aerosol during the lockdown periods, specific PM chemical speciation data (including nitrate, sulfate, and secondary particulate organics) would be needed for 2020 and previous years (for comparison). Such data are not available for the sites of our study at which models tell that decreases in NOx emission would lead to increased secondary aerosol formation. The significance of our work comes from the fact that the lockdown impact on air pollution is evaluated at near 30 sites throughout Europe using the same methodology which shows that conversely to NO2, PM was not strongly impacted by the reduction in human mobility during the 2020 lockdowns in Europe. Beyond NO2 and PM concentrations, we also assess the impact of lockdowns on ozone as an indicator of the oxidative capacity of the troposphere, and on the particle number size distribution and the aerosol light absorption spectrum. To the best of our knowledge, this is the first study to investigate such a set of variables on such a geographical scale.

Specific comments:

1-    The abstract has been modified as suggested (details about the aerosol light absorption spectrum and particle number size distribution analyses have been deleted).

2-    As suggested, the scientific objectives of our study have been listed in the introduction, as follows: (i) determine the impact of the lockdown measures on particulate air pollution at urban and regional background sites across Europe, (ii) explain these results by assessing the impact of the lockdown measures on key gaseous pollutants, on the spectrum of the aerosol light absorption, and on the shape of particle number size distributions (PNSD), and (iii) study the relationship between these impacts and the level of stringency of the lockdown measures across Europe, as estimated from a common indicator for human mobility. The consequences of the lockdown measures could give a hint on the impact of future car exhaust emission reductions on air pollution across Europe.

3-    In the initial version of our manuscript, line 110 actually read "We assumed that the number of particles larger than 15 nm is reasonably insensitive to new particle formation bursts". This sentence has been substituted in the revised version of the manuscript (line 110 – 112) by "The growth of new particles produced during nucleation events also leads to particles in this size range. The number of particles in the size range 15 – 70 nm shall therefore be considered as an upper limit for the number of primary particles". A phrase in Section 3.5.2. which accounts for the fact that the number of particle in the range 15-70 nm can actually be affected by nucleation events, namely "(possibly including nucleation and growth of new particles)", has also been included (lines 400-402).

4-    As stated in Section 3.2 (lines 232 – 235), the differences between the mean observed / expected concentration ratios (as defined in Eq. 3) during the 3 time periods A, D, and P represent our estimates of the lockdown impacts on atmospheric pollution. By using the ratios between the observed and expected concentrations, we could estimate the changes in pollution levels that could not be forecast by the model, namely the changes due to lockdowns. Table 2 caption in the revised version of the manuscript specifies that tabulated values are computed "from the 2020 observed / expected ratios shown in Figures 2 to 8". In more details, the mean impacts of the lockdown implementation and relaxation (across all sites) are estimated for each variable as $average(\langle A_i\rangle/\langle D_i\rangle - 1)$ and $average(\langle D_i\rangle/\langle P_i\rangle - 1)$, respectively, with $i$ representing the site index, and the changes expressed as percentage (0.10 = 10%). Equations 4a and 4b have been included in the revised version of the manuscript.

5-    Daily Obs2019/CAMS2019 ratios are actually used in Eq. 1, as an indicator of the bias between CAMS forecast and actually observed concentrations, which can vary throughout the year. There is no averaging before Eq. 2.

6-    Line 144: "originate" replaced by "originated", as suggested.

7-    Line 180: left bracket inserted as requested.

8-    Line 180: we do not deem it necessary to spell out the country names since this information is irrelevant for the data analysis (e.g., we do not use any national statistics, except for Cyprus). We think that official country codes are sufficient to remove any possible ambiguity, and the map in Figure 1 gives the relevant information regarding the location of the various sites.

9-    Line 218: Section 2.1 has been revised (lines 124-125) to introduce CYSTAT.

10-  Lines 233 and 253: the definition of periods A, D, and P have been deleted, as suggested.

11-  Line 325: to avoid any qualitative statement, this sentence has been revised as "On average, the impact of the lockdown measure implementation and relaxation on $O_3$ concentration at regional background sites was estimated to+17 ± 24 % and +4 ± 5 %, respectively" (lines 330-331).

---

## Author Response (AR2)

We would like to thank again the editor and both referees for the attention they paid to our manuscript ACP-2023-434.

The typo in Line 54 "2017-2029 measurement" has been corrected as "2017-2029 measurement" as rightly suggested by Referee #2.